# A Modified Model Reference Adaptive Controller (M-MRAC) Using an Updated MIT-Rule for the Altitude of a UAV

**Julian Rothe \*, Jasper Zevering, Michael Strohmeier** 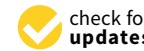 **and Sergio Montenegro \***

Chair of Aerospace Information Technology, Julius-Maximillians-Universität Würzburg, Josef-Martin-Weg, 97074 Würzburg, Germany; info@jasperzevering.de (J.Z.); michael.strohmeier@uni-wuerzburg.de (M.S.)

\* Correspondence: julian.rothe@uni-wuerzburg.de (J.R.); sergio.montenegro@uni-wuerzburg.de (S.M.); Tel.: +49-931-31-88353 (J.R.); +49-931-31-83715 (S.M.)

**Abstract:** Unmanned Aerial Vehicles (UAVs) are playing an increasingly important role in a wide variety of areas and the range of applications increases daily, which can also be seen in the research of the topic. At the University of Wuerzburg drones are to be used in a project, where the aim is to catch possibly dangerous UAVs in mid air using a net, carried by two drones. This very special scenario poses new problems to the control of the drones, so that traditionally used Proportional-Integral-Differential (PID) controllers are no longer sufficient. Therefore a model-based adaption mechanism was chosen to be used to control the altitude of the drones. Though adaption based controllers have been used in the field of drone research before, the existing algorithms had to be modified to work with the special conditions of the altitude control of UAVs. The design and implementation of the modified Model Reference Adaptive Controllers (MRACs) with an updated Massachusetts Institute of Technology (MIT)-rule will be presented in this work. The behavior of the drones with and without the adaption as well as the changes to the original MRAC are then compared in simulation as well as on a real system and show very promising results in further improving the stability of the altitude control of the drones.

**Keywords:** UAV; altitude control; adaptive controller; MRAC; MIT rule

## 1. Introduction

UAVs like quadrocopters have long been non-exceptional in everyday life and, in addition to military and private use, industrial use is also growing steadily. One of the most promising applications for using UAVs in a large number is the transport of different objects, for example, delivering mail to remote places. Since the payload attached to the UAV can not always be known in advance, this increases the need for versatile, adaptable and robust controllers. With the technology becoming more and more part of the economic world, where cost-optimization is often the leading argument when it comes to development, it is no longer sufficient nor even possible to analyze every part of every possible action of a drone and therefore predicting its behavior at any time. Knowing the dynamics of the whole system when building and tuning the controller for a UAV with changing payloads has therefore not only become hard but nearly impossible. Furthermore, with the growing research area of using UAVs in safety and security situations, where robustness and reliability are indispensable, the need to adapt to new situations in a fast and secure way becomes more pressing. One of these projects, named Micro Drone Protection System (MIDRAS) [1], is being developed in parts at the University of Wuerzburg. The project aims to develop a system which is capable of catching UAVs in mid-air in a safe and reliable way. A model of this system can be seen in Figure 1.

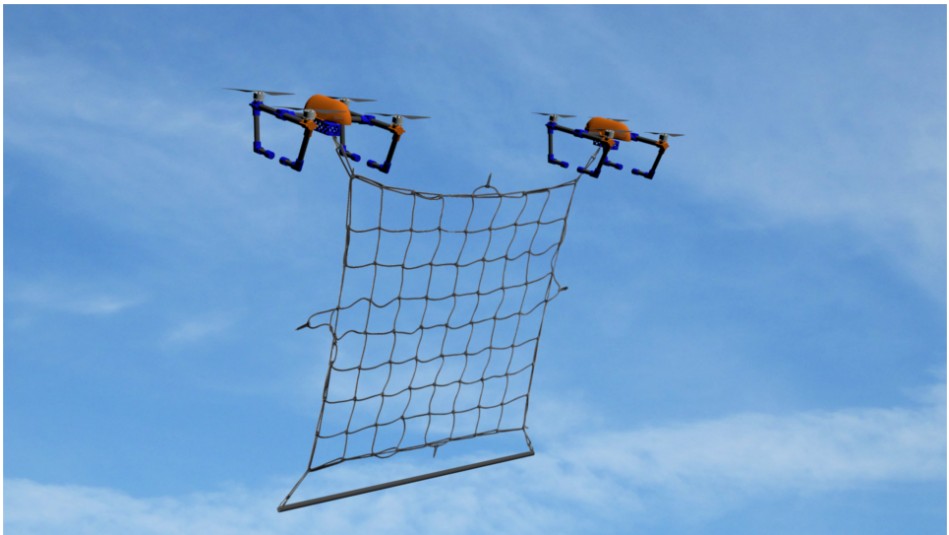

**Figure 1.** Model of the UAV system carrying a net.

As soon as a possible threat is detected, a pair of drones carrying a net will launch and try to catch the target as fast as possible. Since the UAV will be caught at high velocities, the drones carrying the net will have to compensate great forces at the moment of impact. Furthermore, the mass of the drone system will increase significantly with the UAV being tangled in the net. Most UAVs use basic controllers, like PID, to control the attitude and position of the system. These controllers can be perfectly tuned for a known model, but have a hard time to adapt to changing conditions. In our first tests we found that the existing cascaded PID-control-structure for the altitude can handle the high impact and changed mass, but the dynamics of the system change severely, since we can not predict the model after the impact. Therefore, we researched different adaptive control systems to compensate for the loss of dynamics of our system and developed a modified model reference adaptive controller (MMRAC) for the altitude of our drones. These controllers compare the behavior of the UAVs to a model and adapt the output of the controller to minimize the error between model and system. The design, implementation, simulation and evaluation of this controller will be presented in this paper.

## 2. State of the Art

Prof. Karl Johan Åström defines an adaptive controller as follows [2]:

> An adaptive controller is a controller with adjustable parameters and a mechanism of adjusting the parameters.

So when it comes to adaptive controllers, there are three basic considerations to be made:

1. The sort of the underlying controller;
2. The selection of parameters to be adaptively updated;
3. The mechanism updating these parameters.

The sort of the controller is mostly given by the use-case, like PID and Linear–quadratic regulator (LQR). In this work, which basic controller behaves best with an adaptive controller will not be discussed. The most basic idea of an adaptive controller is gain-scheduling. Here, parameters of the controller are adapted by an open-loop controller, like, for example, the level of a kerosene tank. However, this approach can not compensate for changes that can not be predicted or measured. To overcome this shortage, the idea of using a reference model was taken. The simplest way of using a reference model is a high-gain controller, where the desired input $r$ is given to the model, which leads

to reference output $y_m$. This output is used by a closed-loop controller as an input, where the control law is a simple gain $k$. With the transfer function of that closed-loop controller with Plant $P(s)$ being

$$G(s) = \frac{k \cdot P(s)}{1 + k \cdot P(s)},$$ (1)

a high gain $k$ leads to $G(s) = 1$ and therefore $y = y_m$. Of course, in a real system a high gain leads to oscillations, which can be handled, but can also lead to (critical) instabilities as described in [3,4]. One improvement to this is the self-oscillating-controller, which replaces the high gain $k$ by a relay-based gain. This leads directly to maximum-cycle oscillations, even in theory. However, in contrast to high-gain-control, the oscillations are more controllable by a lead-lag filter. Like the high-gain-control, this approach is not usable for sensitive applications like airplanes.

Another improvement of the disadvantages of these controllers leads to MRAC, which will also be used and improved in this work. The scheme can be seen in Figure 2.

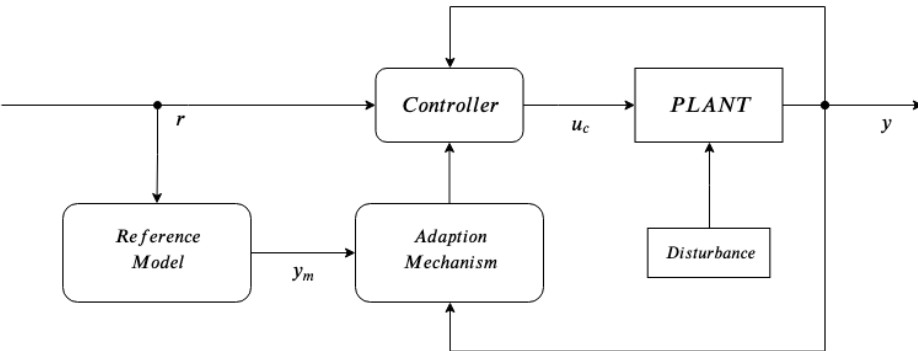

**Figure 2.** Model Reference Adaptive Controller.

The controller of the closed-loop is adapted by an adjustment mechanism, which takes $y_m$ from the reference model as input and tries to adjust the controller output $u_c$. There are two main approaches to this adjustment mechanism: MIT-rule and Stability Proof Rule (SPR)-rule. The MIT-rule adapts a feed-forward gain by the error between the system $y$ and the model $y_m$ and therefore is the so-called Gradient Approach. As this is the approach used by the Modified-MRAC of this work, it will be explained later. The SPR is fully proven with Lyapunov stability. Like the MIT-rule, it uses a feed-forward gain, but the adaptation rule is derived by the state-space model and Lyapunov's Second Theorem [5,6]. The resulting equation for the gain $\theta$ of MIT and SPR rule are:

$$e = y - y_m$$ (2)

$$\dot{\theta}_{MIT} = -\gamma \cdot y_m \cdot e$$ (3)

$$\dot{\theta}_{SPR} = -\gamma \cdot u_c \cdot e$$ (4)

These equations look similar, but differ when it comes to evaluating: Although SPR stability is fully proven and MIT is not, it only works for a smaller class of inputs than the MIT-rule [7].

If the reference system is unknown, there is also an approach of identifying the system model, which is then used by an adaptive controller. There are two main implementations of this, the Self Tuning Regulator (STR) and the Model Identification Adaptive Controller (MIAC) [2,8]. They both share the same approach, but the STR takes the input and output of the plant and estimates so-called "plant parameters", whereas MIAC estimates parameters for a model [2]. They can be applied to a wide range of dynamics. The disadvantage is the hard stability analysis because of the wide variation of parameters. Furthemore, there is a problem with the initialization, which can be solved by an initial state, but this assumes the system to be at least roughly known. Although this seems a more

sophisticated approach, it has already been introduced in 1958 by R. Kalman but due to the lack of enough computational power not seen as usable control-design [9,10]. The last adaptive controller introduced is the stochastic controller, which has come up in the last years. This is a non-heuristic, but purely theoretically-based method, where the system and environment are described as stochastic systems, and the aim is to minimize the loss function. Until now, the only stochastic problem to be considered as solvable is the linear quadratic Gaussian problem, which can be used to tune an LQR-controller. The whole process is highly complex and therefore a very sparse used approach. Regarding the adaptive control of UAVs, there have been several works in the literature. In [11] a basic Lyapunov-based adaptive tracking controller for an underactuated quadrocopter system with unknown parameters, including mass and inertia matrix inter alia, is derived. Tested with simulations, it shows the ability of adaptive controllers of minimizing attitude errors. The idea of an MRAC-controlled UAV regarding its flight stability while the flight dynamics change has already been implemented in [12,13]. Both have their focus on remaining stability while losing the validity of the linear model due to damage to one motor or rotor. These approaches show the ability of MRAC and M-MRAC to handle huge changes in the flight behavior successfully where normal controllers could not have prevented an immediate crash. Nevertheless, both implementations provide improvements for situations for extreme dynamic changes, not for the normal flight behavior and/or small to medium changes in flight dynamics. Therefore the aim of these papers are not the restored flight dynamics, but the save landing and prevention of crashes. M. Schreier derives in [14] a basic MIAC and Lyapunov-based MRAC for Quadrocopter attitude. This was tested in a simulation with inertia-change as a consequence of sudden load-change which were handled by both controllers. A nonlinear disturbance-observer, based on a linear model of a quadrocopter, is taken in [15] to derive an MRAC for compensating external disturbances and was tested successfully in simulations. In [16] a decentralized MIT-based MRAC-attitude-controller for quadrocopters is introduced. It stabilizes hover maneuvers and trajectory tracking under the addition of Gaussian white noise of measurements and external perturbations.

These works focused on either a good flight behavior or keeping controllability when having considerable changes in the dynamics. Most times the controller was designed with the adaptation mechanism braided into it, where an approach for an adaptive controller attached to the regular controller without changing it was not followed up. Furthermore, most work in the literature is either focusing only on the algorithms or only using adaptive controllers with UAVs in simulations and never test the results on a real system. This can result in unexpected errors, when testing the algorithms on real drones. As for our project the underlying MRAC resulted in an unstable flight behavior and had to be modified to be used for the altitude control. These are the main focuses of the research on M-MRAC of this work, which will be further discussed in the next section.

## 3. Design of the Adaption Mechanism

The controller presented in this work was specifically designed for the UAVs used in the project MIDRAS described in Section 1. Therefore the requirements for the design of the adaptive altitude controller are the following:

1.　No or only minimal changes to the current cascaded PID controller;
2.　Adaption without endangering the robustness or stability of the system;
3.　Fast adaption to an abrupt change of the forces, most likely due to a change of mass;
4.　Keeping the change of the dynamics of the system as minimal as possible when conditions alter.

These requirements lead to two main conclusions for the adaptive mechanism of the MRAC, which will be discussed in more detail.

### 3.1. M-MRAC

Because the changes to the current controller should be minimal, the adaption to changes of the system is achieved by multiplying the adaptive gain $\theta$ to the output of the controller $u_c$.

$$u(t) = \theta \cdot u_c(t). \tag{5}$$

This method of adapting can be used to adapt to slower changes but can become unstable for abrupt changes to the system. Therefore we implemented a so called modified MRAC (M-MRAC) as presented in [17–19]. This modification of MRAC adds a PID-controller to the adjustment mechanism. This results in the following adjusted output of the controller.

$$u(t) = \theta \cdot u_c(t) + \left( k_p \cdot e + k_i \cdot \int e \ dt + k_d \cdot \dot{e} \right) \tag{6}$$

where $k_p, k_i, k_d$ are the parameters of the PID controller and $e$ is the error between the plant $y$ and the reference model $y_m$ as seen in Figure 2.

$$e = y - y_m \tag{7}$$

Abrupt changes to the system, i.e., as the target hits the net, are absorbed by this additive PID controller, while the change of the mass and dynamics will still be handled by the MIT gain $\theta$.

### 3.2. Modified MIT Rule

Since the dynamics of the altitude control for a UAV are very high compared to other more stable systems, we use the MIT-Rule to adjust our adaptive gain $\theta$. The MIT-rule was developed at the MIT, hence its name. In our work we use the MIT-rule as presented in [20–22], but with a new modification, which will be described in the following equations. First, there is the definition of a cost function

$$J(\theta) = \frac{1}{2} \cdot e^2 \tag{8}$$

with $\theta$ being the adjustment parameter and $e$ the error between the plant $y$ and the reference model $y_m$, as for the M-MRAC. Since the aim of the adaption mechanism is to minimize the cost function, the change of $\theta$, $\dot{\theta}$, has to be kept in the negative gradient of $J$.

$$\dot{\theta} = -\gamma \cdot \frac{\delta J}{\delta \theta} \tag{9}$$

where $\gamma$ is an adjustable quantity that indicates the speed of the adaption. With (8) and defining the partial derivative $\varphi = \frac{\delta e}{\delta \theta}$ this becomes

$$\dot{\theta} = -\gamma \cdot e \cdot \frac{\delta e}{\delta \theta} = -\gamma \cdot e \cdot \varphi. \tag{10}$$

Similar to [20], we now define the transfer function of our system to be $G(s)$, an unknown influence on the system as parameter $K$ and a known modeled influence as $K_0$. Then we can rewrite (7) in the frequency domain as:

$$E(s) = K \cdot G(s) \cdot U(s) - K_0 \cdot G(s) \cdot U_c(s) \tag{11}$$

with the control law described in Equation (5). Therefore the partial differentiationcan be formulated with substitutions as:

$$\frac{\delta E(s)}{\delta \theta} = K \cdot G(s) \cdot U_c(s) = \frac{K}{K_0} \cdot K_0 \cdot G(s) \cdot U_c(s). \tag{12}$$

With these equations, and using $\gamma'$ as a new adjustable parameter to combine the constants $\gamma, K, K_0$, we can now write (10) as

$$\dot{\theta} = -\gamma \cdot e \cdot \varphi = -\gamma \cdot e \cdot \frac{K}{K_0} \cdot y_m = -\gamma' \cdot e \cdot y_m \tag{13}$$

This equation to change the adjustment parameter $\theta$ is used throughout the literature as described in [20–22]. Though this will work for most systems, there is a problem in the special case of the reference model $y_m = 0$. This will result in an unchanging adjustment parameter $\theta$ even if the error between model and system $|e| > 0$ and therefore in an unstable system behavior.

$$\dot{\theta} = -\gamma' \cdot e \cdot y_m = -\gamma' \cdot e \cdot 0 = 0. \tag{14}$$

In the context of this work, this can happen if the altitude of the drone is only controlled by the inner vertical velocity controller and set to be $v_{des} = 0\frac{m}{s}$. A change in mass will now result in an error between model and plant, but the adjustment parameter will not be affected since the reference model will stay $y_m = 0\frac{m}{s}$. As a consequence the error between the reference model and the plant will not be corrected and there is no benefit of the MRAC. Therefore we modified the MIT-rule by formula (12) to be depended on $y$ with the following substitution using (5)

$$\frac{\delta E(s)}{\delta \theta} = K \cdot G(s) \cdot U_c(s) = K \cdot G(s) \cdot U_c(s) \cdot \frac{\theta}{\theta} = \frac{K \cdot G(s) \cdot U(s)}{\theta}. \tag{15}$$

and

$$\dot{\theta} = -\gamma \cdot e \cdot \frac{y}{\theta}. \tag{16}$$

Further [20] use a normalization of the MIT-rule to overcome the problem of instability with large inputs. To do this they define $\dot{\theta}$ as

$$\dot{\theta} = \frac{-\gamma \cdot e \cdot \varphi}{\alpha + \varphi^2} \tag{17}$$

where $\alpha$ is introduced to remove the problem of the possible division by zero if $\varphi$ gets too small. With our new definition of $\varphi$ this becomes

$$\dot{\theta} = \frac{-\gamma \cdot e \cdot \frac{y}{\theta}}{\alpha + \left(\frac{y}{\theta}\right)^2} \tag{18}$$

## 4. Implementation

Without the proposed adaptive controller, the altitude of the existing UAV is controlled by a cascaded PID structure. The outer controller uses the desired altitude $p_{z_{des}}$ and the current altitude $p_z$ to set the desired velocity $v_{z_{des}}$, which is used in combination with the current vertical velocity $v_z$ by the inner controller to calculate the needed control output $u_c$. This output is added to a constant HeightGas $u_0$, which is the gas needed to counter the gravitational forces on the UAV. The position $p_z$ and velocity $v_z$ of the UAV are calculated by fusing different sensors in a Kalman Filter. Figure 3 shows the previous existing implemented control structure for the altitude of the UAV.

The added components of the modified MRAC with its adaption mechanisms to the existing control structure are highlighted. As described in Section 3 there are two mechanisms to adjust the performance of the system to follow a reference model. The MIT rule uses the current vertical velocity $v_z$ and the reference velocity $v_{z_{ref}}$ to adjust $\theta$, which is then multiplied to the output of the existing controller $u_c$. The additive PID controller also uses $v_z$ and $v_{z_{ref}}$ to calculate a control output $u_{add}$,

which is added to the existing control output to adjust to abrupt changes of the system behavior. This leads to an updated adjusted output of the controller.

$$u(t) = \theta \cdot u_c(t) + u_{add}(t) + u_0 \tag{19}$$

In the implementation the reference model is described by a second order transfer function with two poles. It is important to state, that the reference model used by the MRAC should not be understand as a close representation of the real UAV system, but rather a desired behavior of the UAV, closely designed in comparison to the real system.

$$G(s) = \frac{b}{s^2 + c_1 \cdot s + c_2} \tag{20}$$

The parameters of the transfer function were found by analyzing the step response of the system with different parametersand optimizing them to have no overshoot and a satisfying settle time, while still being close to the observed behavior of the drone.

$$G(s) = \frac{90}{s^2 + 17 \cdot s + 90}. \tag{21}$$

This transfer function $G(s)$ uses the output $v_{z_{des}}$ of the outer PID controller for the altitude of the UAV to calculate the output of the model or reference velocity $y_m = v_{z_{ref}}$. With this, both adaption mechanisms adjust the output of the system, as described in Section 3. The only notable practical issue, when implementing the discussed design is, that a control output $u_c = 0$ gives no acceleration to the system, because the HeightGas $u_0$ is additionally added to the output. Since one of the requirements was to keep the changes to the existing controller as minimal as possible, this problem was solved by shifting the output of the controller to the positive, when using the modified MRAC. To do this we first add the maximum output of the gas $gas_{max} = 100$ to the output, which is modified by $\theta$, before subtracting the same value again.

$$u(t) = \theta \cdot (u_c(t) + 100) - 100 + u_{add}(t) + u_0 \tag{22}$$

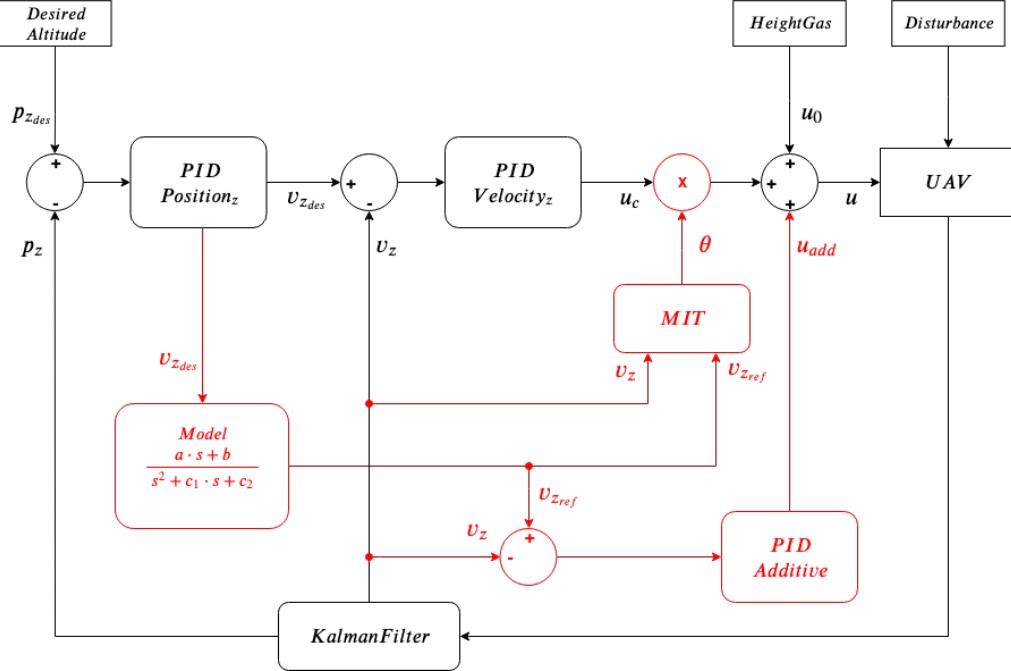

**Figure 3.** Diagram of the implemented modified MRAC.

## 5. Simulation

Before testing the proposed new approach on a real UAV, the algorithms were simulated in Simulink [23]. Therefore a model of the cascaded PID control structure for the altitude and vertical velocity was created. To simulate the physics, a simplified model of the motion of the UAV in the z-axis was derived by measuring the produced force of the motors for given PWM-signals and the response of the UAV. Combined with the mass of the later used UAV of $m_{drone} = 215$ g the simplified model was accurate enough to be used for the test of the stability and reliability of the algorithms.

### 5.1. Comparison to a Standard Controller

In the first simulation we compared the performance of the modified MRAC with the proposed changes to the performance of the standard cascaded controller. Therefore the simulated outer PID-controller for the altitude was set to hold the height at $z = 2$ m. After reaching this position, a mass of $m_{weight} = 100$ g, which equals to almost half the weight of the UAV, was attached to the simulated model at $t = 20$ s and released at $t = 40$ s for both controllers. The resulting height for both controllers can be seen in Figure 4.

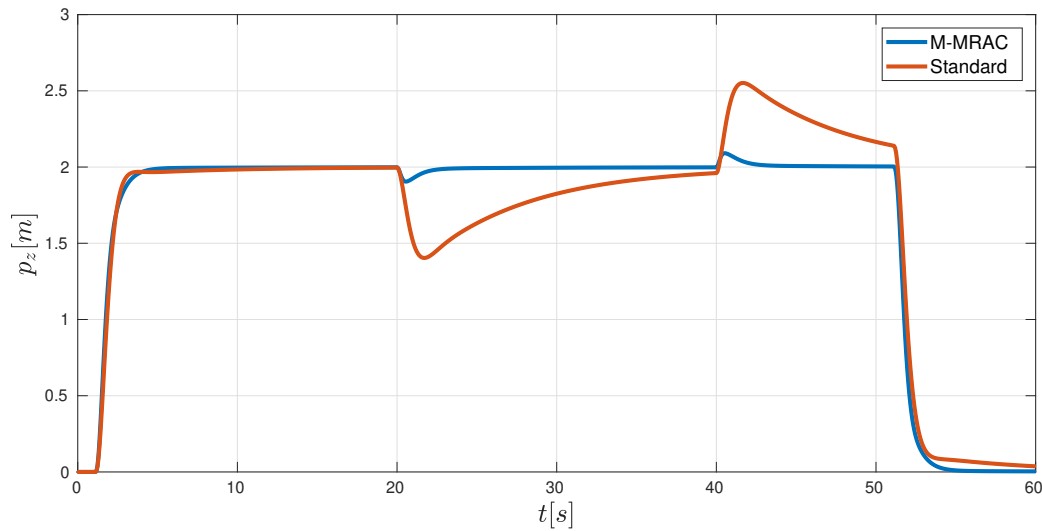

**Figure 4.** Simulation of the UAV performing a position-controlled flight whilst changing the weight.

The difference in the performance of the two simulated control structures can clearly be seen in the plot of the altitude of both simulations. When attaching and releasing the additional mass at $t = 20$ s and $t = 40$ s respectively, the modified MRAC only shows a peak overshoot of $e_{max} = 0.1$ m and returns to the desired position almost immediately. In comparison the standard controller has a peak overshoot of $e_{max} = 0.6$ m and is not able to return to the original height in the given time span of $\Delta t = 20$ s before being commanded to land again.

### 5.2. Benefit of the Additive PID-Controller

Next we show the benefit of the additive PID controller in the adjustment mechanism to the system. For this the same simulation as in the previous evaluation was run again with and without the additive PID-controller in the adjustment mechanism for the MRAC.

Figure 5 shows the position and the velocity in the z-axis for both controllers, when adding and releasing the mass to the model. While both controllers react fast and with little overshoot in the position $p_z$ to the change in mass when compared to the standard controller in Figure 4, the additive PID controller smooths the response of the controller by reacting immediately to the change in weight. The benefit can especially be seen in the velocity $v_z$ of the UAVs. The normal approach of MRAC results in a overshooting and oscillating velocity $v_z$, which could result in an unstable UAV. Therefore

a higher adaption gain $\gamma$ can be used without risking oscillations in the system, which results in a faster adaption to changes in the system.

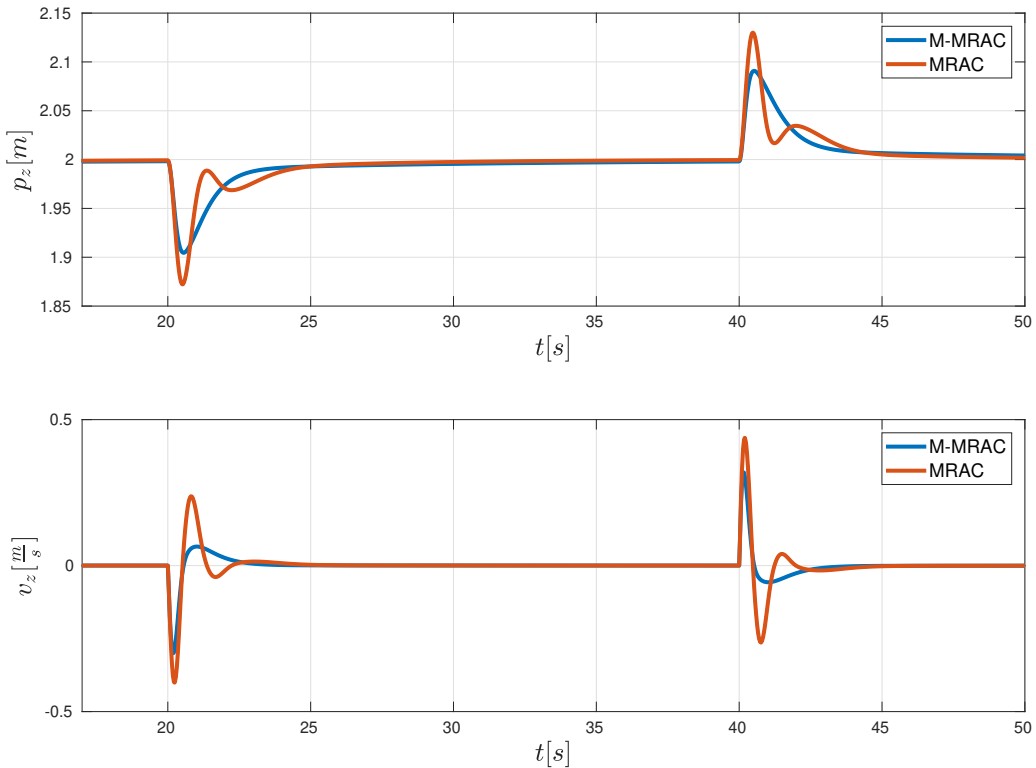

**Figure 5.** Simulation of the difference of the additive PID controller.

### 5.3. Benefit of the Modified MIT-Rule

Lastly, the proposed modifications to the MIT-rule with MRAC are compared to the original version. As described in Section 3 the motivation for the proposed changes is the problem, which occurs when the reference value is set to $y_m = 0$. In our implementation (see Section 4) the reference value $y_m$ is described by the reference velocity $v_{z_{ref}}$, which is in turn given by a transfer function $G(s)$ using the desired velocity $v_{z_{des}}$. To better show the resulting error of these special conditions, the models were simulated for two different scenarios $v_{z_{des}} = 2 \frac{m}{s}$ and $v_{z_{des}} = 0 \frac{m}{s}$. Both models where only controlled by the inner PID controller for the vertical velocity. Again at $t = 20$ s a mass of $m = 100$ g was attached to the simulated model and released at $t = 40$ s for both controllers.

In Figure 6 the resulting velocities and gain $\theta$ are shown for the scenario $v_{z_{des}} = 2 \frac{m}{s}$. For this, both controller structures give almost identical results as expected. The change of the weight results in a drop respectively increase in the velocity. This error $v_{err}$ results in a change in the gain $\theta$ and is therefore corrected after a short period of time.

However, when setting the desired velocity $v_{z_{des}} = 0 \frac{m}{s}$ as in the second scenario seen in Figure 7, the results of both versions differ. In the standard version the algorithm for calculating the change of the gain $\theta$ is

$$\dot{\theta} = -\gamma' \cdot e \cdot y_m = -\gamma' \cdot v_{z_{err}} \cdot v_{z_{ref}} = 0 \tag{23}$$

Therefore $\theta$ is not changing, even if the error $e > 0$, resulting in an unchanged behavior of the model. The error of the velocity $v_{z_{err}}$ after changing the weight of the model is only being compensated by the I-part of the PID controller. As seen in Figure 7 the velocity $v_z$ can not even be corrected in the given time span of $\Delta t = 20$ s. On contrary the proposed new modified MIT-rule derives the change of $\theta$ with

$$\dot{\theta} = -\gamma \cdot e \cdot \frac{y}{\theta} = -\gamma \cdot v_{z_{err}} \cdot \frac{v_z}{\theta} \neq 0. \tag{24}$$

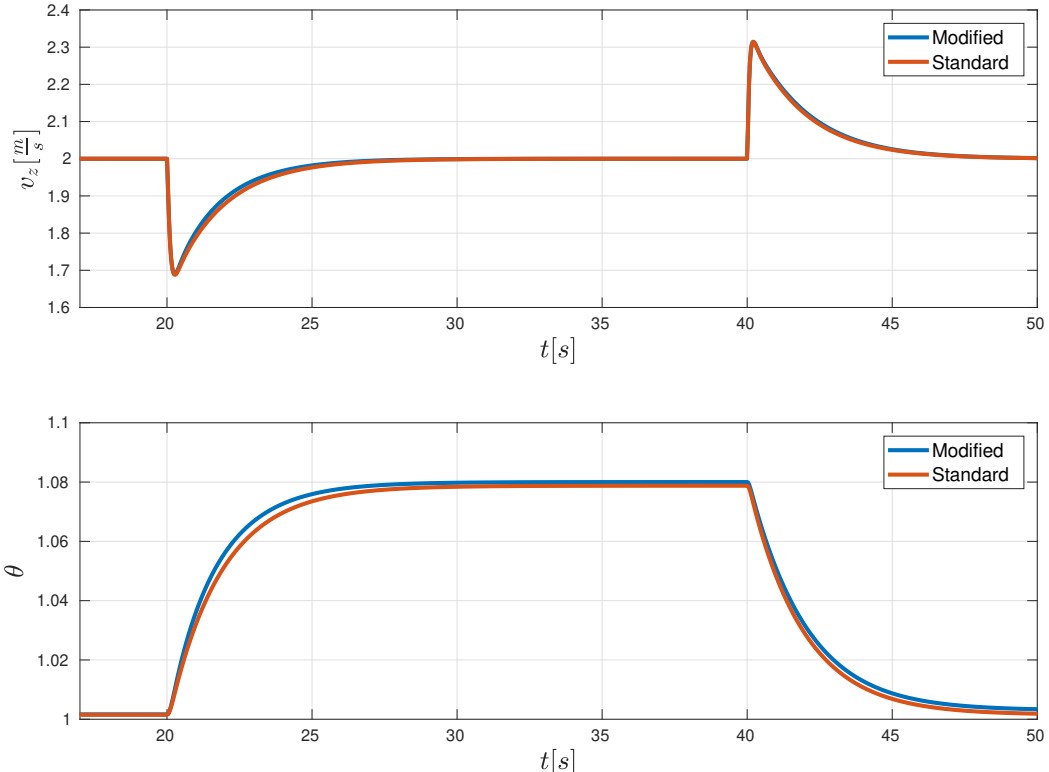

**Figure 6.** Simulation of the difference of the additive PID controller.

Hence it reacts to the change of mass by adjusting $\theta$ and is able to alter the velocity to $v_z = 0 \, \frac{\text{m}}{\text{s}}$ in the same time period as in the scenario with $v_z = 2 \, \frac{\text{m}}{\text{s}}$.

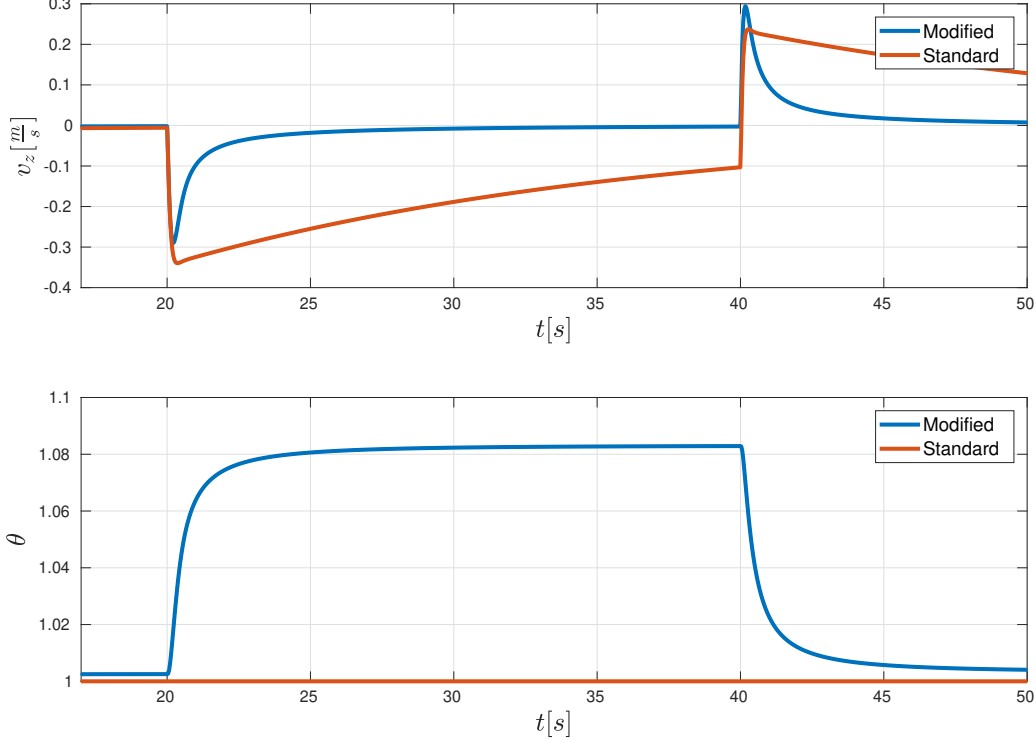

**Figure 7.** Simulation of the difference of the additive PID controller.

## 6. Real System

After the successful simulation of the modified MRAC, the algorithms were tested on a real system. As described before, the modeled UAV used for the simulation in the previous Section 5 was designed based on the drone used for the later evaluation with the real system.

The micro-UAV as seen in Figure 8 was developed at the University of Wuerzburg [24]. The autopilot software on the flight controller runs on a cortex M4-processer using the Real-time Onboard Dependable Operating System (RODOS) [25]. The UAV has a mass of $m = 215$ g with a footprint of 15 cm $\times$ 15 cm and the following, for this work related, components:

- Processor: STMicroelectronics: STM32F407VG;
- Accelerometer: STMicroelectronics: LSM9DS1;
- Electronic Speed Controller (ESC): Sunrise: Cicada-30A-4in1;
- Motors: Emax RS1106 II.

For the indoor localization and evaluation of the performance, an optical tracking system (OptiTrack [26]) was used to provide centimeter accurate positions. These are fused with the data of the other sensors in a 19 state Kalman filter [27] to estimate the position and orientation of the drone. All following comparisons where run on the same drone. In contrast to the simulations in the previous section, the attached weight was not just a modeled weight difference, but a real mass in form of a clamp, which was attached by hand to the drone at a given time. Therefore, slight variations in the time are possible due to human error.

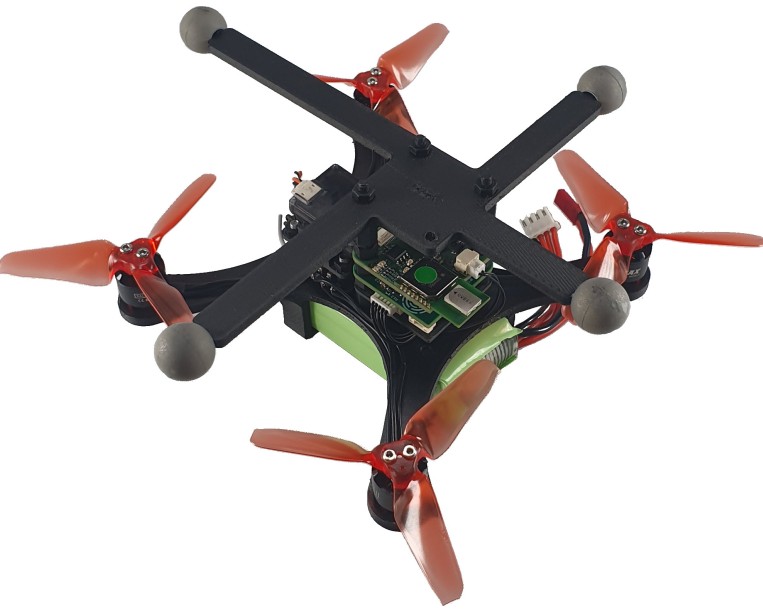

**Figure 8.** Micro-UAV with optical markers used for the evaluation.

### 6.1. Comparison to a Standard Controller

To compare the performance of the used M-MRAC with the previous existing control structure, both controllers were commanded to perform the same procedure as in Section 5. First, the altitude controller was set to reach and hold an altitude of $z = 2$ m. Next, a mass of $m = 118$ g, which is more than half the weight of the UAV, was attached to the drone at $t = 34$ s and released at $t = 64$ s.

Figure 9 shows the altitude of the UAV for both controllers. As expected from the simulations, the modified MRAC improves the performance of the system strongly. When attaching and releasing the clamp the adaptive controller has a peak overshoot of $e_{max} = 0.45$ m, while the overshoot of the standard controller is more than three times higher $e_{max} = 1.5$ m. Furthermore, the time to reach the desired altitude of $z = 2$ m after the change of mass is reduced significantly by the MRAC.

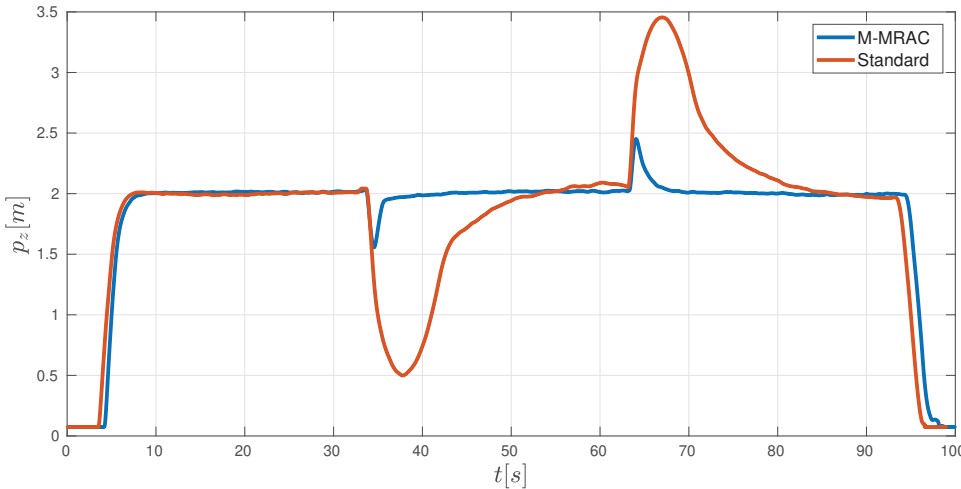

**Figure 9.** Real system comparison of the UAV performing a position controlled flight whilst changing the weight.

### 6.2. Benefit of the Additive PID-Controller

For the difference between the approach with and without the additive PID-controller in the adjustment mechanism, the same procedure as in the previous evaluation was run again. The plots of position and velocity in Figure 10 show only the relevant part of the experiment, when attaching and releasing the mass. Unfortunately, since the weight was manually attached at a given time, there is a small offset in the timing. It is still evident to see, that the additive part is very beneficial to the adjustment mechanism as it smooths the response of the UAV. Without it, the system tends to oscillate after the mass of the system is changed, while the settling time stays approximately the same. The oscillations can be seen in the plots of position $p_z$ and the velocity $v_z$. Especially the resulting behavior, when releasing the mass seems to be unstable when only using the standard MRAC. The added PID controller dampens the great changes of the model and therefore keeps the system more reliable.

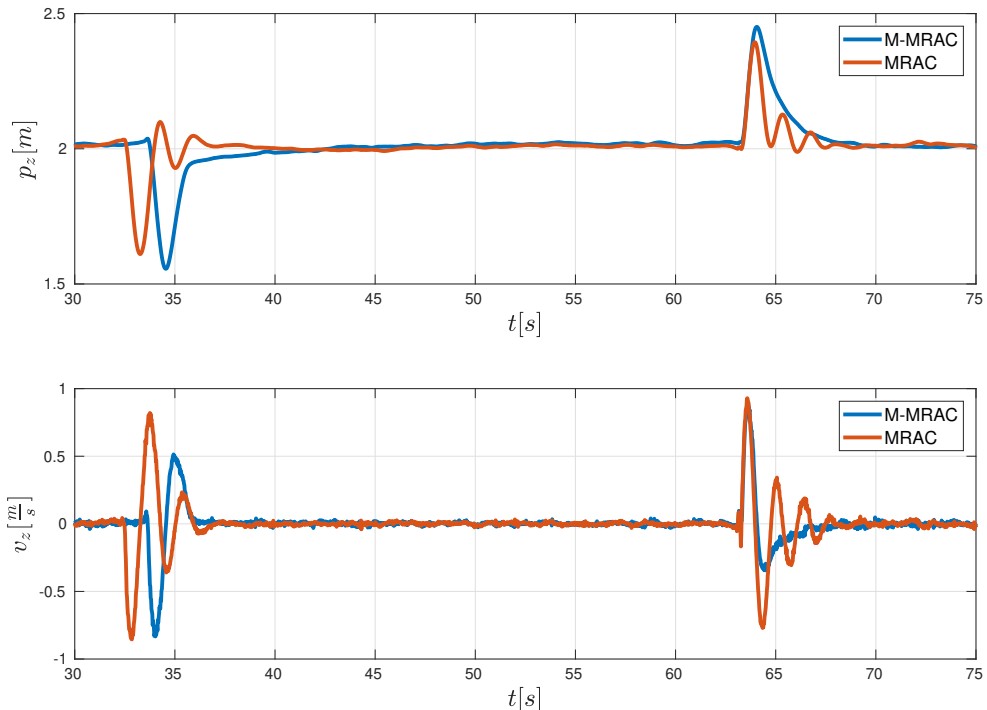

**Figure 10.** Difference of the additive PID controller for the real system.

### 6.3. Benefit of the Modified MIT-Rule

In the last evaluation we show the main improvement of the new approach compared to the known MRAC with MIT-Rule. Therefore the UAV was commanded to a height of $z = 2$ m and the mass was attached to the UAV. After the system had adapted to the new weight, the controller was changed to the inner velocity-PID controller with a $v_{z_{des}} = 0 \frac{m}{s}$ and the mass was released at $t = 25$ s.

As seen in Figure 11 the proposed new modified MRAC immediately adapts to the change of the weight by adjusting the gain and therefore reaches its commanded velocity after a small overshoot. In comparison to that, the old controller does not change the gain, because of the problems in Section 3 described. This results in a long settling time of the velocity, since the velocity-PID-controller is not adapted by the MIT-mechanism, but only adjusts its I-part to compensate for the change in mass. Even though the difference might not seem to be too big, this resulted the UAV to change its position $\Delta p_z = 2.4$ m, while the change of position with the new modified version of MRAC was only $\Delta p_z = 0.1$ m.

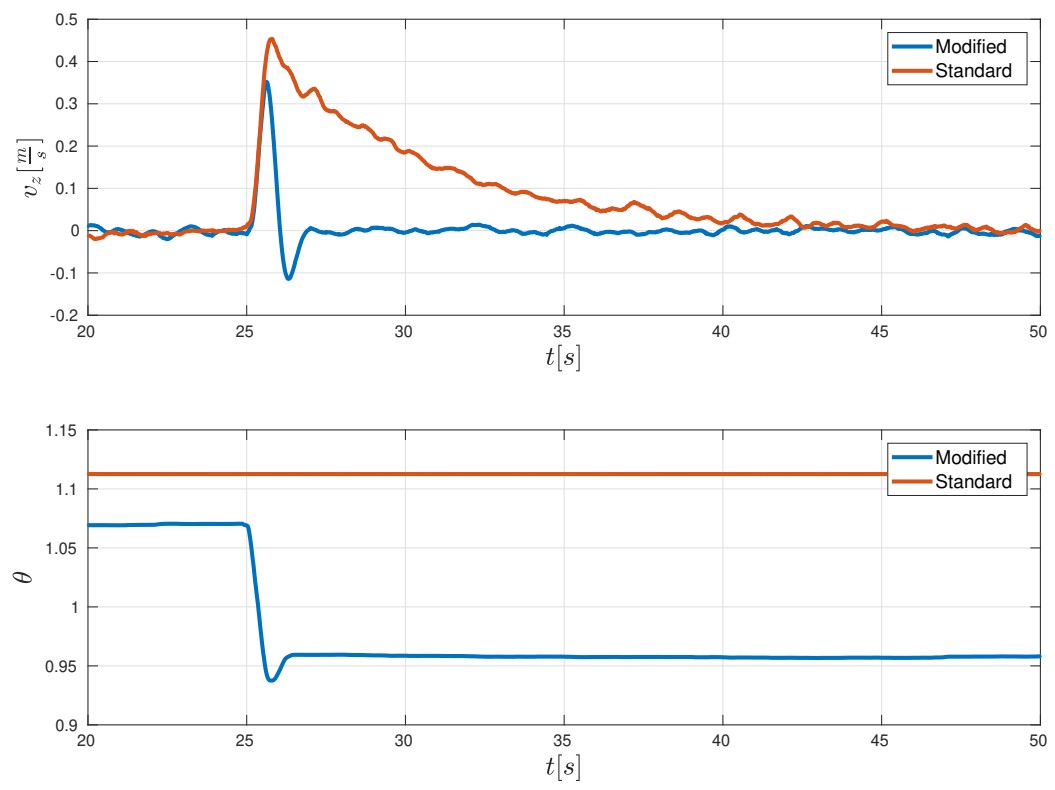

**Figure 11.** Difference of the new and old approach for M-MRAC.

## 7. Conclusions

In this paper we showed the design, implementation and evaluation of a modified-Model Reference Adaptive Controller (M-MRAC) for the altitude of a UAV using a new modified version of the MIT-rule. The evaluation of changing the mass of a UAV in flight, show promising results both in the simulation as on a real system. Furthermore the modifications improved the existing algorithms in the special case of $y_{ref} = 0$. In the next steps this algorithm needs to be tested on different UAVs and finally in the underlying project MIDRAS, where it will help the performance when two drones carry a net in a formation flight.

**Author Contributions:** J.R. designed the algorithms, implemented the presented approach, conducted the experiments and wrote this manuscript. J.Z. implemented the presented approach and wrote the state of the art M.S. and S.M. directed the research and gave critical feedback. All authors have read and agreed to the published version of the manuscript.

**Funding:** This research has been funded by the Federal Ministry of Education and Research of Germany in the framework of MIDRAS (project number 13N14315).

**Conflicts of Interest:** The authors declare no conflict of interest.

## Abbreviations

The following abbreviations are used in this manuscript:

| | |
|---|---|
| **ESC** | Electronic Speed Controller |
| **LQR** | Linear–quadratic regulator |
| **MIAC** | Model Identification Adaptive Controller |
| **MIDRAS** | Micro Drone Protection System |
| **MIT** | Massachusetts Institute of Technology |
| **MRAC** | Model Reference Adaptive Controller |
| **PID** | Proportional-Integral-Differential |
| **RODOS** | Real-time Onboard Dependable Operating System |
| **SPR** | Stability Proof Rule |
| **STR** | Self Tuning Regulator |
| **UAV** | Unmanned Aerial Vehicle |

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
