# Peer review of "A Modified Model Reference Adaptive Controller (M-MRAC) Using an Updated MIT-Rule for the Altitude of a UAV"

_electronics, doi:10.3390/electronics9071104_

Round 1
Reviewer 1 Report
I found the paper interesting and quite well written. I suggest a minor amendments to improve its readability.
In eq. (6) You propose to add full PID controller? Why full? Is it necessary to apply all its elements?
How did you obtained eq. (13)?
On basis of what have you assumed eq. (15)?
in eq. (22) you add 100*(theta-1), it is not clear why? why such a way?
Simulations, could you additionally present how theata is changing?
Author Response
Thank you very much for taking the time to read our paper and give your constructive feedback. It really helped us, to further improve the quality of our submission. We hope that we addressed all of your suggested changes and comments to your complete satisfaction.
To highlight the changes we used the LaTeX package “changes” (see https://www.ctan.org/tex-archive/macros/latex/contrib/changes). A list of all changes can be found at the end of the document with the respective line numbers.
This highlighting of the changes also resulted in a slightly shifted formatting of the document, which we ask to excuse at this point of the reviewing progress, as it will be resolved before final submission.
In eq. (6) You propose to add full PID controller? Why full? Is it necessary to apply all its elements?
Yes we use a full PID-controller for M-MRAC, as it is also suggested in the literature. We tested different values for the PID and found, that the best results with a full PID-controller.
How did you obtained eq. (13)?
Equation 13 is a direct result of using Equation 12 to substitute delta. To further clarify this an additional step was added to Equation 12 to show how it can be arranged in dependency of y.
On basis of what have you assumed eq. (15)?
Equation 15 shows the same step to substitute delta as in equation 13, but instead of having a dependency to y_m, we rearrange equation 12 to be dependend on y. To further clarify this, equation 15 was replaced to have the same style as equation 12 and show how it can be arranged in dependency of y using the equations presented beforehand.
in eq. (22) you add 100*(theta-1), it is not clear why? why such a way?
As stated this is a special problem to our system, since the adaption mechanism is adapting the control output without the heightGas. Therefore an output of u(t) = 0 will lead to no acceleration, since the heightGas is set to counter the gravitational forces. By first adding and then substracting 100 to the output of the modified MRAC, we shift our output to the positive, so that we will have an impact on the behaviour of the UAV. The number 100 is chosen, since this is the maximum gas output of the controller. This was added to the paper to clarify the statement. (see line 197)
Simulations, could you additionally present how theata is changing?
We discuss the change of \theta compared to the presented modified aproach to MRAC in detail in chapter 5.3 as well as in chapter 6.3. Here the change of theta can be observed for different velocities when attaching and releasing a mass to a drone in simulation as well as on the real system.
Showing the change of theta for chapter 5.1 is not relevant, since the system without an adaption has no adaption gain theta. Chapter 5.2 highlights the difference of M-MRAC and MRAC, so that the change in theta is also not relevant here. We hope that this clarifies the question enough.
Reviewer 2 Report
The task of two UAV drones is to catch possibly dangerous UAV in mid air using a net.
- The authors claim that the traditionally used PID controllers are no longer sufficient. Please state the reason. The environment is mild air, it should state the reason for the ineffectiveness of PID in mild environment.
2. The introduction part is weak. More topic on Modified Model Reference Adaptive Control should be given.
3. The UAV model is very simple, which is only second-order system in (3)- (4). Please verify the accuracy of your method in experiment.
4. The definitions of many variables are missing, such as \gammar
5. Since Model Reference Adaptive Control is quite mature now, please state your distingguished contributions.
6. The experiment validation is required to show the merits of your method.
Author Response
Thank you very much for taking the time to read our paper and give your constructive feedback. It really helped us, to further improve the quality of our submission. We hope that we addressed all of your suggested changes and comments to your complete satisfaction.
To highlight the changes we used the LaTeX package “changes” (see https://www.ctan.org/tex-archive/macros/latex/contrib/changes). A list of all changes can be found at the end of the document with the respective line numbers.
This highlighting of the changes also resulted in a slightly shifted formatting of the document, which we ask to excuse at this point of the reviewing progress, as it will be resolved before final submission.
1. The authors claim that the traditionally used PID controllers are no longer sufficient. Please state the reason. The environment is mild air, it should state the reason for the ineffectiveness of PID in mild environment.
As we state in the paper, we are working on a project, where two drones try to catch another drone with a net. Therefore the load of the UAVs changes rapidly. Traditionally used PID-controllers can adapt to changing of conditions, but first of all take a long time to this and second tend to loose the dynamic that they had, since the PID parameters are tuned for a known system. Since we do neither know the weight, nor the velocities at the point of impact, a PID-controller is not sufficient to be used in this scenario. A sentence to clarify this was added (line 35). This is also what we tested and showed in simulations (Chapter 5.1) and on a real system (Chapter 6.1).
It is not completely clear, what the is meant by “mild air” or how “mild air” influences the flight of UAVs.
2. The introduction part is weak. More topic on Modified Model Reference Adaptive Control should be given.
A sentence highlighting the effect of model reference adaptive controllers was added in line 35 and 38.
The Introduction focused on the reason, why a traditionally used PID controller is not sufficient by explaining the problem of our project in detail. Therefore most of the introduction to Modified Model Reference Adaptive Controllers is given in Chaper 2 – State of the art, since we wanted to give a broad overview of the topic and show, that such controllers have only been used in rare cases with UAVs.
3. The UAV model is very simple, which is only second-order system in (3)- (4). Please verify the accuracy of your method in experiment.
The model of the UAV shown in Figure 3 is the real model for the altitude control of the real UAVs. This second order system is used on all research projects with UAVs at our department and has been in use for several years. The adaption mechanism added to this second-order system was tested in simulations and on the real system in several hours of flight time. A few of the experiments are shown in Chapter 5 and 6. As it can be seen, when comparing results of the simulation with the real system, the predicted output by the simulations using this model came very close to the output of the real system.
We used this model to find a suitable transfer function as stated in Chapter 4 (Equation 21). The phrasing of how the parameters were decided was rewritten, as this as maybe led to wrong conclusions on the accuracy of the model. The transfer function is not the real model of the UAV, but should rather be seen as a desired behavior of the UAV.
4. The definitions of many variables are missing, such as \gammar
Definitions of variables were added in lines 66, 67, 131, 150.
5. Since Model Reference Adaptive Control is quite mature now, please state your distingguished contributions.
The first distinguished contribution is that we use the M-MRAC to not only keep the system stable, after the model has changed, but also to keep the dynamics the same. As we state in the Chapter 1, 2 and 3, Model reference adaptive controllers have been used with UAVs before, but in all the cited works it was only used to land the aircraft in a safe manner, after the model of the system had changed. Most of the works do only work with simulations and never test or compare their results with a real system. This was added to the paper in line 108).
The second mayor distinguished contribution is the modification of the change of the adaption parameter as stated at the end of chapter 3.2. With the existing version of MRAC used throughout the literature, an error did not always result in a change of the adaption parameter, therefore resulting in an unstable system. This was also shown in simulation in 5.3 and on the real system in 6.3.
6. The experiment validation is required to show the merits of your method.
The experiment validation comparing the improved version to the “standard MRAC” was further highlighted where it was not yet described in detail in the chapters 5.2, 6.2, 6.3.
Reviewer 3 Report
Comments:
To solve the controlling problem that two drones catch possibly dangerous UAV in mid air using a net, this paper designed a modified MRAC with an updated MIT-rule and presented the implementation. The simulation results and real system measurements show very promising results in stability of the altitude control.
Major suggestions:
- More disadvantages of original MRAC and the novelties of this paper should be given to support the research necessity.
- Most of the equations already exist in the other literatures, and the authors may strengthen their own works and improvements such as equation (15)-(16).
- The simulation results and real system test show greater performance than the standard controller, but the other improved MRACs in the referenced literatures may also have better performance than standard controller and the authors should compare the results with the other improved MRACs to show the advantage of the proposed modified MRAC.
Minor suggestions:
- Some abbreviations in this paper should be given in the first place they are used, such as PID.
- Some paragraphs have no first line indent, e.g., Line 24 in page 1, Line 33 in page 2, etc.
Author Response
Thank you very much for taking the time to read our paper and give your constructive feedback. It really helped us, to further improve the quality of our submission. We hope that we addressed all of your suggested changes and comments to your complete satisfaction.
To highlight the changes we used the LaTeX package “changes” (see https://www.ctan.org/tex-archive/macros/latex/contrib/changes). A list of all changes can be found at the end of the document with the respective line numbers.
This highlighting of the changes also resulted in a slightly shifted formatting of the document, which we ask to excuse at this point of the reviewing progress, as it will be resolved before final submission.
1.More disadvantages of original MRAC and the novelties of this paper should be given to support the research necessity.
2.Most of the equations already exist in the other literatures, and the authors may strengthen their own works and improvements such as equation (15)-(16).
Further detailed explanation of the benefit of this work were added in lines 35, 108, 155, 158.
Many works in the literature that have been using MRAC with UAVs focus either on the algorithms of the adaption mechanism or only on the simulation. There are only a few works in general on MRAC that show the use of the controller on a real system, which is most of the time very challenging to transfer the knowledge of the simulations to a real system. Therefore this work partly focuses on showing the results of the controller comparing the simulations with the behavior of a real drone, which is shown in detail in chapters 5 and 6.
Equation 15 was rearranged to further clarify the changes made to the original MRAC system.
3. The simulation results and real system test show greater performance than the standard controller, but the other improved MRACs in the referenced literatures may also have better performance than standard controller and the authors should compare the results with the other improved MRACs to show the advantage of the proposed modified MRAC.
The improved performance compared to the “standard MRAC” was further highlighted where it was not yet described in detail in the chapters 5.2, 6.2, 6.3.
The only known improvement to MRAC called modified MRAC or MMRAC is also used in this work and the results of using the added PID-controller is shown in chapters 5.2 and 6.2. Other improvements, i.e. using a model identification adaptive controller (MIAC) were tested but resulted in unstable behavior of the UAV, when changing the model significantly by changing the weight. Since the results of this research are to be used in a critical system for drone defense, we decided no to go into further research.
The improvement of the proposed modified change of the adaption parameter is shown in detail in chapters 5.3 and 6.3 and were further described.
1. Some abbreviations in this paper should be given in the first place they are used, such as PID.
Abbreviations given in the first place of usage, see lines 5 and 10
2. Some paragraphs have no first line indent, e.g., Line 24 in page 1, Line 33 in page 2, etc.
Blank lines removed, see lines 24 and 33.
Round 2
Reviewer 2 Report
I suggest to accept it after English improvement.
Reviewer 3 Report
Our comments have been well addressed. In my opinion, the current version of the manuscript is more readable and useful to the potential readers.